# Tetranucleotide and Low Microsatellite Instability Are Inversely Associated with the CpG Island Methylator Phenotype in Colorectal Cancer

**DOI:** 10.3390/cancers13143529

**Published:** 2021-07-14

**Authors:** Sabine Meessen, Nicola Currey, Zeenat Jahan, Hannah W. Parker, Mark A. Jenkins, Daniel D. Buchanan, John L. Hopper, Eva Segelov, Jane E. Dahlstrom, Maija R. J. Kohonen-Corish

**Affiliations:** 1Garvan Institute of Medical Research, Sydney, NSW 2010, Australia; Sabine.meessen@medma.uni-heidelberg.de (S.M.); Nicola.Dyson@newcastle.ac.uk (N.C.); 2The Woolcock Institute of Medical Research, The University of Sydney, Sydney, NSW 2037, Australia; zeenat.jahan@sydney.edu.au (Z.J.); hannah.w.parker@student.uts.edu.au (H.W.P.); 3Faculty of Science, University of Technology Sydney, Sydney, NSW 2007, Australia; 4Centre for Epidemiology and Biostatistics, Melbourne School of Population and Global Health, University of Melbourne, Melbourne, VIC 3000, Australia; m.jenkins@unimelb.edu.au (M.A.J.); j.hopper@unimelb.edu.au (J.L.H.); 5Department of Clinical Pathology, University of Melbourne, Melbourne, VIC 3010, Australia; daniel.buchanan@unimelb.edu.au; 6University of Melbourne Centre for Cancer Research, University of Melbourne, Melbourne, VIC 3010, Australia; 7Genomic Medicine and Family Cancer Clinic, Royal Melbourne Hospital, Melbourne, VIC 3010, Australia; 8Department of Oncology, Monash University and Monash Health, Melbourne, VIC 3168, Australia; eva.segelov@monash.edu; 9ACT Pathology, The Canberra Hospital and Australian National University Medical School, Canberra, ACT 2605, Australia; Jane.E.Dahlstrom@act.gov.au; 10Microbiome Research Centre, St George & Sutherland Clinical School, UNSW Sydney, Sydney, NSW 2217, Australia; 11School of Medicine, Western Sydney University, Sydney, NSW 2751, Australia

**Keywords:** EMAST, MSI-H, MSI-L, CIMP, colorectal cancer, *CDKN2A*, *MCC*, MSH3

## Abstract

**Simple Summary:**

A type of DNA mismatch repair defect known as “elevated microsatellite alterations at selected tetranucleotide repeats” (EMAST) is found across many different cancers. Tetranucleotide microsatellite instability, which is caused by *MSH3* mismatch repair gene/protein loss-of-function, shares a molecular basis with “low microsatellite instability” (MSI-L) in colorectal cancer. Tetranucleotide microsatellite instability is also a byproduct of “high microsatellite instability” (MSI-H) that arises from deficiency of mismatch repair due to *MSH2,* *MSH6*, *MLH1* or *PMS2* gene alterations. *MSH3*-related EMAST is emerging as a biomarker of poor prognosis in colorectal cancer and needs to be clearly differentiated from MSI-H. Here, we show that tumours with non-MSI-H-related EMAST or MSI-L rarely show concordant promoter methylation of multiple marker genes. Colorectal tumours that are positive for a single (1/5) tetranucleotide repeat marker are an important subset of the EMAST spectrum.

**Abstract:**

*MSH3* gene or protein deficiency or loss-of-function in colorectal cancer can cause a DNA mismatch repair defect known as “elevated microsatellite alterations at selected tetranucleotide repeats” (EMAST). A high percentage of MSI-H tumors exhibit EMAST, while MSI-L is also linked with EMAST. However, the distribution of CpG island methylator phenotype (CIMP) within the EMAST spectrum is not known. Five tetranucleotide repeat and five MSI markers were used to classify 100 sporadic colorectal tumours for EMAST, MSI-H and MSI-L according to the number of unstable markers detected. Promoter methylation was determined using methylation-specific PCR for *MSH3*, *MCC*, *CDKN2A* (p16) and five CIMP marker genes. EMAST was found in 55% of sporadic colorectal carcinomas. Carcinomas with only one positive marker (EMAST-1/5, 26%) were associated with advanced tumour stage, increased lymph node metastasis, MSI-L and lack of CIMP-H. EMAST-2/5 (16%) carcinomas displayed some methylation but MSI was rare. Carcinomas with **≥**3 positive EMAST markers (13%) were more likely to have a proximal colon location and be MSI-H and CIMP-H. Our study suggests that EMAST/MSI-L is a valuable prognostic and predictive marker for colorectal carcinomas that do not display the high methylation phenotype CIMP-H.

## 1. Introduction

Microsatellite instability (MSI) in cancer is caused by deficient repair of insertion–deletion loop mismatches, which occur during DNA replication in tandem repeat sequences throughout the genome [1,2]. MSI-H is caused by DNA mismatch repair (MMR) deficiency as a result of alterations in the *MLH1*, *MSH2*, *MSH6* or *PMS2* genes. Another form of MSI is “elevated microsatellite alterations at selected tetranucleotide repeats” (EMAST) that has been identified in up to 60% of colorectal cancers [3,4,5,6], but no standard marker panel has been adopted yet for its detection. EMAST is caused by a deficiency or dysfunction of MMR involving the *MSH3* gene or protein [5,7]. Loss of *MSH3* expression or function can occur through multiple mechanisms, such as gene mutations, transcriptional or post-transcriptional repression, or cellular mislocalisation of the MSH3 protein during colon inflammation [4,5,8,9,10]. The *MSH3* gene is silenced through promoter methylation in gastric cancer [11], but the relevance of promoter methylation for the generation of EMAST has not been reported yet in colorectal cancer.

The MSH3/MSH2 (MutSβ) heterodimer binds to DNA and initiates the repair of insertion–deletion loop mismatches of 2–4 nucleotides, while MSH6/MSH2 (MutSα) is more specific for mismatches of single base-pairs and insertion–deletion loop mismatches with 1–2 nucleotides [4]. The MLH1/PMS2 (MutLα) complex has a coordinator role in MMR. EMAST is caused by MutSβ dysfunction due to a defective MSH3 or MSH2. However, MSH2 deficiency disables both MutSβ and MutSα and can cause concurrent EMAST and MSI-H. Similarly, MutLα dysfunction due to a MLH1 or PMS2 defect can cause both EMAST and MSI-H. Therefore, there is an obligatory overlap of tetranucleotide repeat instability and MSI-H colorectal cancers, but MSH3-related EMAST is linked with the MSI-L phenotype [4,6,12,13].

The CpG island methylator phenotype (CIMP) is an important predictive factor in colorectal cancers involving concordant methylation silencing of multiple tumour suppressor genes [14]. Patients with CIMP-H cancers do not receive a benefit from 5-FU-based chemotherapy [15,16]. CIMP-high (H) is defined as hypermethylation of at least 3/5 markers while CIMP-low (L) is defined as hypermethylation of 1–2 out of 5 markers. CIMP-H is associated with *MLH1* promoter methylation and the resulting MSI-H [17]. However, the extent of the association of CIMP with EMAST/non-MSI-H colorectal cancers has not been determined. Using the Weisenberger et al. panel of five markers [17], CIMP-H is found in ~11% and CIMP-L in ~20% of colorectal carcinomas. In addition to the five CIMP markers, genes that show concordant methylation in colorectal carcinoma include *CDKN2A* (p16) and *MCC* [18]. *CDKN2A* (p16) is an important cell cycle regulator that is silenced in ~30% of colorectal cancers [19]. The *MCC* gene is emerging as a regulator of the DNA damage response and epithelial cell–cell adhesion and is silenced by methylation in 44% and 24% of colon and rectal carcinomas, respectively [20,21,22,23]. We have shown that loss of *Mcc* down-regulates DNA repair genes, including *Msh3*, in a mouse model of inflammation-associated colon cancer [21].

Therefore, the aim of this study was to determine the association of EMAST with (i) clinicopathological features in colorectal carcinoma, (ii) MSI, (iii) CIMP and (iv) methylation of selected genes *MSH3*, *MCC* and *CDKN2A* (p16). We also wanted to investigate if these associations could help classify EMAST subtypes according to the number of markers present in each tumour.

## 2. Materials and Methods

### 2.1. Patient Cohorts

DNA was extracted from fresh frozen primary colorectal tumours and matched normal colon specimens (cohort 1) obtained from cancer resections (2000–2003). Thirty four percent of the patients received subsequent chemotherapy. This cohort has been previously analysed for clinicopathological and molecular variables from formalin-fixed paraffin embedded (FFPE) or fresh frozen tissue [20,24]. Clinical, pathological and follow-up data for all patients were obtained from the South Western Sydney Clinical Cancer Registry.

Further FFPE tumour DNA specimens and associated biomarker data were obtained from The Jeremy Jass Memorial Pathology Bank and the Australasian Colorectal Cancer Family Registry (ACCFR) housed at University of Melbourne (cohort 2) [25,26]. These included 22 Lynch syndrome associated carcinomas, with verified MMR gene deficiency (15 MLH1, 6 MSH2, 1 PMS2) and MSI-H status, and their matching control DNA extracted from peripheral blood [27].

The UNSW Biorepository at Mark Wainwright Analytical Centre, UNSW Sydney provided DNA specimens extracted from 13 colitis-associated colorectal carcinoma (fresh frozen) (cohort 3) [28]. Blood samples were not available and, therefore, EMAST-positivity could only be determined relative to the matching non-carcinoma inflamed tissue. The Australian Pancreatic Cancer Genome Initiative donated DNA extracted from 33 pancreatic cancers (fresh frozen) and their non-neoplastic tissue or blood control samples (cohort 4) [29,30]. Two of the tumours were selected for the study because of their known MSI-H status while 31 tumours were assigned randomly. No clinical follow-up data were available for cohorts 2–4.

### 2.2. Extraction of DNA and RNA from Tissue

DNA and RNA were simultaneously extracted from frozen colorectal tissue specimens using the Allprep DNA/RNA Mini Kit (Qiagen, Hilden, Germany), following homogenisation (Polytron, Kinematica, Littau, Switzerland).

### 2.3. Analysis of EMAST and MSI

There is no standardised reference panel available yet to define EMAST, but several suitable tetranucleotide markers have been identified. Tetranucleotide repeat markers D9S242, D20S82, L17686, UT5320, MYCL1 were selected from a previous publication [6]. PCR amplification of the markers was combined into multiplex reactions, using 1U of MyTaq polymerase and 1× MyTaq buffer (Bioline, London, UK) in each reaction. The first multiplex contained primer mixes for D9S242 and L17686 at 0.4 µM, and the second combined primers for D20S82, UT5320 and MYCL1 at 0.24 µM. Thermocycling was carried out on a Mastercycler Nexus (Eppendorf, Hamburg, Germany), with an initial denaturation of 1 min at 95 °C, followed by 35 cycles of 30 s at 95 °C, 15 s at 54–57 °C and 90 s at 72 °C, and a final extension of 10 min at 72 °C.

MSI analysis was performed using the reference panel BAT25, BAT26, D2S123, D5S346, D17S250 [3]. Cancers were classified as MSI-H if ≥2/5 of the marker genes were unstable and MSI-L when one marker was unstable. The PCRs were multiplexed, the first combining 0.2 µM BAT25 with 0.8 µM BAT26, and the second comprising 0.3 µM D2S123, 0.3 µM D5S346 and 0.5 µM D17S250. Amplification was carried out by 10 min denaturation at 95 °C, followed by 40 cycles of 45 s at 95 °C, 45 s at 55 °C and 60 s at 68 °C, and a final extension of 10 min at 68 °C.

Capillary electrophoresis of amplified products was carried out on the ABI3130XL (Thermo Fisher Scientific, Waltham, MA, USA), and data analysis performed using GeneMapper software (Thermo Fisher Scientific, Waltham, MA, USA).

### 2.4. Analysis of Gene Methylation and Gene Expression

Bisulphite conversion of tumour and matching control DNA was carried out using the Epitect Bisulfite Kit (Qiagen, Hilden, Germany). *MSH3* methylation status was determined by methylation-specific PCR on bisulphite-treated DNA. PCR reactions specific for methylated and unmethylated *MSH3* [11] were performed, with universally methylated and unmethylated DNA (Merck Millipore, Burlington, MA, USA) used as positive and negative controls. Amplified PCR products were run on agarose gels and visualised with the E-Box imager (Vilber Lourmat, Collégien, France). *MCC* and *CDKN2A* (p16) methylation and CIMP were analysed as previously described [18,19,31]. Cancers were classified as either CIMP-H if ≥3/5 of the marker genes showed methylation, CIMP-L when 1–2/5 markers were positive or CIMP-negative when 0/5 markers were positive.

cDNA was prepared from total RNA (or mRNA) using the Quantitect Reverse Transcription Kit (Qiagen, Hilden, Germany), following the manufacturer’s protocol. *MSH3* expression was analysed by quantitative RT-PCR using Taqman assays (Thermo Fisher Scientific, Waltham, MA, USA). *MSH3* (hs00989003_m1) was normalised to *GAPDH* (hs99999905_m1). Assays were carried out in triplicate on the ABI7900HT (Thermo Fisher Scientific) and the ΔΔCT method was used for data analysis.

### 2.5. Statistical Analysis

Survival curves were created using the Kaplan–Meier method and compared with the log-rank (Mantel–Cox) test (GraphPad Prism, San Diego, CA, USA). Overall survival was calculated from the date of diagnosis to the date of death due to any cause. Patients who were alive or lost to follow-up were censored. Fisher’s exact test and chi square was used to evaluate the significance of differences in contingency tables. The level for statistical significance was set at ≤0.05.

### 2.6. Analysis of The Cancer Genome Atlas (TCGA) Dataset

The TCGA PanCancer Atlas dataset containing comprehensive integrated molecular analyses was accessed through the cBioPortal for Cancer Genomics platform (https://www.cbioportal.org; accessed on 30 June 2021) [32,33,34]. The cohort of 594 Colorectal Adenocarcinoma patients was selected for data display [34]. The following genomic profiles were selected for the *MSH3* gene: (i) mutations, (ii) putative copy-number alterations and (iii) mRNA expression z-scores relative to normal tissue samples. This returned data for 524 samples. The same analysis was performed for the TCGA PanCancer Atlas Pancreatic Adenocarcinoma cohort [34], except that mRNA expression z-scores were calculated relative to diploid samples (corresponding normal tissue data were not available). The graphs and statistical data were obtained using the function “Plots” and “Color samples by MSI MANTIS score” [35]. The Illumina Methylation450K data for colorectal and pancreatic cancers were obtained using the SMART App (http://www.bioinfo-zs.com/smartapp; accessed on 23 December 2020) [36].

## 3. Results

### 3.1. Patterns of EMAST Marker Positivity Are Different between Colorectal and Pancreatic Carcinomas

A panel of five tetranucleotide markers was selected from a previous publication to detect EMAST [6]. We first determined the prevalence of instability for each of these five EMAST markers across the four cohorts of carcinomas where MSI-H is found at a high frequency (100%, Lynch syndrome, cohort 2), intermediate (13–15%, sporadic colorectal and colitis-associated carcinomas, cohorts 1 and 3) or low frequency (sporadic pancreatic carcinomas, cohort 4). Instability for at least one of these EMAST markers was found in 100% of the 22 Lynch syndrome-associated colorectal carcinomas, 55% (55/100) of sporadic colorectal, 62% (8/13) of colitis-associated, and 74% (23/31) of sporadic non-MSI-H pancreatic carcinoma. Positivity for each of the five markers was evenly distributed in the 55 EMAST-positive sporadic colorectal carcinomas (19–24%). In contrast, the L17686 tetranucleotide marker was unstable in the majority of the colitis-associated (5/8) and sporadic pancreatic (21/23) carcinomas with any EMAST-positive markers. Representative images of the EMAST markers are shown in Figure 1.

### 3.2. EMAST≥3/5 Is Associated with MSI-H Across Different Cancer Types

Concurrent instability of two or more EMAST markers was found in 95% of Lynch syndrome (21/22), 29–31% of sporadic colorectal (29/100) or colitis-associated (4/13) carcinomas and 19% of the pancreatic carcinomas (6/31). All but two of the MSI-H sporadic colorectal carcinomas (cohort 1) had **≥**3–5 unstable EMAST markers, which was rare in non-MSI-H tumours (Figure 2a, Table 1). Conversely, non-MSI-H colorectal carcinomas showed 1–2 EMAST-positive markers (47%) or were EMAST-negative (51%) (Figure 2a, Table 1). Most of the Lynch syndrome colorectal carcinomas (18/22), 2/2 MSI-H colitis-associated and 2/2 MSI-H pancreatic cancers showed instability with 3–5 EMAST markers.

We then examined the EMAST subsets separately in the sporadic colorectal cancer cohort, EMAST **≥** 3/5 (three or more EMAST-positive markers), EMAST-2/5 (two markers) and EMAST-1/5 (a single marker).

### 3.3. Presence of a Single EMAST Marker Is Associated with MSI-L and Lymph Node Metastases But Not Location in Sporadic Colorectal Carcinoma

We determined the relationship between the number of EMAST markers, MSI-L, tumour stage and location in the sporadic colorectal tumours (Table 1, Figure 2b–d). MSI-L was enriched in EMAST-1/5 tumours (9/26, 35%; Table 1, Table 2, Figure 2a,b). Dukes stage 3 and 4 cancers were also increased in EMAST-1/5 tumours (18/26, 69%), which was due to an association with increased lymph node metastasis (Table 1, Figure 2c,d). In general, these carcinomas were not poorly differentiated and there was no significant difference in location of EMAST-1/5 carcinomas along the length of the large bowel. Similar associations were seen for MSI-L sporadic colorectal carcinomas (Table 2). This suggests that carcinomas with MSI-L and/or a single EMAST positive marker are more likely to be of a higher stage with evidence of lymph node metastases at the time of detection. In survival analysis, there was no difference in overall survival between the EMAST subsets if the EMAST-2/5 group was combined with either EMAST-1/5 (Figure 2f) or EMAST-3/5 (Figure 2g). However, patients with EMAST-1/5 tumours showed significantly shorter overall survival when compared with all other patients as a single group (Figure 2h).

There was no significant association of the number of EMAST positive markers with MSI-L in the colitis-associated carcinomas, where MSI-L was detected at 38% frequency. Only three pancreatic specimens showed MSI-L.

### 3.4. EMAST-1/5 Colorectal Cancers Show Only Scattered Methylation

We next determined the relationship of EMAST with CIMP and methylation of *CDKN2A* (25%) and *MCC* (29%). EMAST **≥**3/5 was strongly associated with CIMP-H and concurrent *MCC*/*CDKN2A* promoter methylation in sporadic colorectal cancer. In contrast, EMAST-1/5 carcinomas showed the lowest frequency of these markers (Table 1, Figure 2a,e). MSI-H was also strongly associated with CIMP-H in sporadic colorectal cancers, as expected, and with concurrent *CDKN2A*/*MCC* methylation. CIMP-H, *MCC* and *CDKN2A* methylation was absent or rare in MSI-L carcinomas (Table 2, Figure 2a). These data are consistent with our previous findings on *CDKN2A* [31] and suggest that carcinomas with MSI-L and/or a single EMAST positive marker are less likely to display high-level gene promoter methylation.

We reported previously that *MCC* and *CDKN2A* promoter methylation are frequently found in the same colon carcinoma specimens [18]. Here, the same strong association was observed in the colon but not in rectal carcinomas. *MCC* and *CDKN2A* methylation co-occurred in 57% (16/28) of the carcinomas positive for either marker in the colon but only 25% (2/8) in the rectum/rectosigmoid region.

The relationship of EMAST with the methylation markers was not studied in pancreatic or colitis-associated carcinomas.

### 3.5. Presence of 1–2 EMAST Markers Is Not Associated with Down-Regulation of MSH3 mRNA Expression

Defective MSH3 function or loss of gene expression causes EMAST and can occur through multiple mechanisms, but the role of *MSH3* gene silencing has not yet been investigated in colorectal cancer. We analysed *MSH3* promoter hypermethylation using methylation-specific primers [11] and RNA expression by qPCR. No promoter hypermethylation was evident in our specimens (data not shown). About 41% (39/96) of the tumours displayed at least a 2-fold reduction in *MSH3* transcript levels compared to matching normal tissue; of these, 13% (12/96) showed >4-fold reduction. Down-regulation of *MSH3* expression was associated with the presence of 3–5 EMAST markers but not with 1–2 positive markers (Table 3). Down-regulation of *MSH3* was also more frequent in MSI-H or CIMP-H cancers but the strongest association was observed with *MCC* methylation (*p* = 0.0001, Table 3).

### 3.6. MSH3 Gene Mutations Are Rare But Copy Number Alterations Are More Common in Colorectal and Pancreatic Carcinoma

To evaluate *MSH3* gene alteration and expression in a larger cohort, we analysed publicly available data from ~3000 colorectal cancer patients using cBioPortal for Cancer Genomics [32,33]. *MSH3* mutations have been detected in 1–6% in various cohorts; about one quarter of these are truncating mutations, the rest are missense mutations. Two (2/524) truncating mutations are reported for colorectal carcinoma in the TCGA PanCancer Atlas cohort [34]. *MSH3* copy number alterations occur in another ~32% of colorectal cancers; one copy is deleted in 24% (128/524), both copies in 0.2% (1/524) and copy gain is found in 8% (41/524) (Figure 3a) [34]. *MSH3* copy number alterations show very little overlap with MSI-H that is mostly confined to *MSH3* diploid cancers [35] (Figure 3b), but tetranucleotide repeat alterations have not yet been captured in the publicly available data.

*MSH3* mRNA expression levels vary greatly in the TCGA colorectal cohort but can be considered low in 17% (87/524) of the tumours, when compared with the mean expression in normal tissue specimens (Figure 3a). There is a weak correlation between copy number alterations and mRNA expression (Pearson *r* = 0.35, *p* < 0.0001). The lowest level of *MSH3* mRNA expression is seen in a subset of diploid tumours, including some MSI-H cancers (Figure 3b). Similar frequencies of *MSH3* mutations and copy number alterations are found for pancreatic adenocarcinomas, but no MSI-H is evident (Figure 3c,d). The *MSH3* gene promoter is not hypermethylated in colorectal or pancreatic carcinomas (Illumina Methylation450K CpG island probes cg11646734, cg16401290, cg14865507, cg07526021 and cg18200270) [34,36].

The TCGA data [34] indicate that *MSH3* mutations, deep deletion or low mRNA expression are not frequent enough to explain the generation of all the observed EMAST in non-MSI-H colorectal or pancreatic carcinomas. Furthermore, our data suggest that down-regulation of MSH3 mRNA does not correlate with the presence of 1–2 EMAST-positive markers in colorectal cancer.

## 4. Discussion

This study has shown that the high methylation phenotype CIMP-H and concurrent *MCC*/*CDKN2A* methylation are rare in colorectal carcinomas that are positive for only one EMAST marker or MSI-L but more common in cancers with ≥2 markers. We previously reported in another cohort that *CDKN2A* (p16) methylation was significantly less frequent in MSI-L colon tumours compared to MSI-H or MSS tumours and that MSI-L tumours were associated with poor survival [31]. Since then, evidence has been strengthening that both MSI-L or EMAST/non-MSI-H are markers of poor prognosis in colorectal carcinoma and share a mechanistic basis [7,12,37,38]. Here, we show that the poor prognosis is likely due to increased lymph node metastasis, which is associated with the presence of one positive EMAST or MSI marker. Our data suggest poorer overall survival for the EMAST-1/5 group, which incorporates most of the MSI-L tumours in the cohort. This supports and extends the findings of Garcia et al., who found that EMAST/non-MSI-H cancers grouped with MSI-L were associated with shorter recurrence-free survival than cancers that were MSI-H or microsatellite stable [12].

Here, we also show that the *MSH3* gene promoter is not methylated in colorectal cancer. Our data and the TCGA data indicate that tumour levels of *MSH3* mRNA expression vary greatly. Down-regulation of *MSH3* mRNA expression is associated with EMAST ≥ 3/5 and MSI-H but not with the EMAST/MSI-L phenotype. This is consistent with the finding that EMAST can be caused by MSH3 protein dysfunction, which occurs as a result of displacement of MSH3 from the nucleus to the cytoplasm. This can be induced by exposing colon cancer cells to IL6 or hydroxen peroxide in vitro and is thought to occur as a result of inflammation in sporadic and inflammatory bowel disease-associated cancers [5,39].

It remains to be determined whether *MSH3* copy number alterations, found in 32% of the TCGA colorectal and 27% of the pancreatic cancer cohort (Figure 3a,c), are associated with the EMAST/non-MSI-H phenotype. MSI-H is rare in colorectal cancers with *MSH3* copy number alterations but common in diploid cancers (Figure 3b). *MSH3* copy number alterations reflect chromosomal instability (CIN), which is known to show inverse correlation with CIMP-H [40]. Here, we found that CIMP-H was rare in cancers with only one positive EMAST marker. Aberrant MSH3 protein expression and loss of heterozygosity of *MSH3* markers have been reported in MSI-L cancers [38]. This may indicate a wider association of CIN with EMAST/MSI-L.

Genes that are silenced by methylation in colorectal carcinoma and associated with CIMP-H include many tumour suppressor genes [17]. Our data show that the EMAST-2/5 and high-level gene methylation show a small overlap in non-MSI-H cancers. The TCGA data show only low correlation between *MSH3* copy number alterations and mRNA expression, indicating that multiple factors regulate mRNA expression. *MSH3* mRNA down-regulation was strongly associated with *MCC* methylation in our cohort, which was not observed for *CDKN2A* methylation. This is consistent with our data from a mouse model of inflammation-associated colorectal cancer where deletion of *Mcc* caused down-regulation of multiple DNA repair genes in the inflamed tissue [21]. *MCC* methylation is also associated with more invasive colorectal cancers [20]. It is possible that methylation or alterations of non-MMR genes modify the EMAST phenotype spectrum, but this needs further investigation.

In this study, we used five tetranucleotide repeat markers, D9S242, D20S82, L17686, UT5320 and MYCL1, that were selected from a previous publication [6]. Two of these markers, L17686 and UT5320, have not been commonly used in other recent studies on EMAST. Here, we detected a higher frequency of EMAST in pancreatic cancer (74%) than previously reported (45%) [41], which was due to the L17686 marker. The L17686 marker was also common in the colitis-associated cancer specimens. Inflammation plays a critical role in the tumorigenesis of both pancreatic and colitis-associated cancer, and IL-6 is an important inflammatory cytokine that promotes pathogenesis in both conditions [10,42,43]. This may suggest that L17686 could be a marker that is particularly sensitive to inflammation-induced EMAST/MSI-L. The frequency of EMAST in colitis-associated cancers was 68%, which is comparable to a previously reported frequency of 71% [10]. However, since our control specimens were from matched inflamed tissue, which could be EMAST-positive, the frequency of EMAST may have been higher. Positivity of a single marker was common in pancreatic cancers (48%, 15/31) using the EMAST marker panel, but less frequent (10%, 3/31) using the MSI marker panel. This highlights the possible differences between cancer types and the need to standardise EMAST marker panels and the nomenclature, possibly incorporating MSI and EMAST markers, as was suggested by Garcia et al. [12] and Raeker et al. [13].

Our study highlights the need to include colorectal or pancreatic cancers that are positive for only one (1/5) tetranucleotide repeat marker as an important subgroup of the EMAST spectrum. In some recent studies, EMAST was defined as the presence of ≥2/5 tetranucleotide markers [44,45,46]. This definition results in a significant overlap with MSI-H and potentially excludes tumours that are associated with increased lymph node metastasis and lack of CIMP-H. The frequency of the EMAST-1/5 group could be inferred in at least two of the studies, as 23% [45] and 28% [46] of MSS tumours, which is comparable to our study.

*MSH3*-associated EMAST/MSI-L is a potentially valuable predictive biomarker for colorectal and other cancers, e.g., pancreatic and non-small cell lung carcinomas [41,47]. Apart from MMR, MSH3 has another function in the homologous recombination (HR) repair of DNA double-strand breaks that occur during carcinogenesis [48]. As a consequence, carcinomas with loss of MSH3 function may have a defect in HR and might respond better to targeted therapies with PARP and/or DNA-PKc inhibitors. These therapies have a synergistic effect based on the synthetic lethality principle [48]. Therefore, it is important to develop biomarkers that can identify the HR defective subset of carcinomas that may be responsive to these or other cancer-targeted drugs.

## 5. Conclusions

This study has identified new features of tetranucleotide instability that are clinically relevant in sporadic colorectal carcinomas. In particular, cancers with only one positive EMAST marker or MSI-L are associated with lack of CIMP-H and an increased likelihood of lymph node metastasis. EMAST-1/5 and MSI-L may be valuable surrogate markers for MSH3 dysfunction. Due to the dual function of MSH3 in both MMR and HR, EMAST/MSI-L cancers could be potentially responsive to therapies that exploit the concept of synthetic lethality.

## Figures and Tables

**Figure 1 cancers-13-03529-f001:**
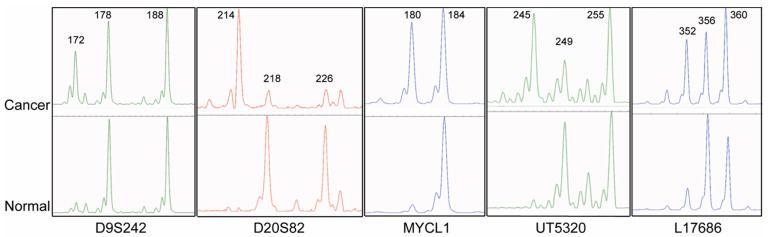
Representative images of EMAST detection using the tetranucleotide repeat markers D9S242, D20S82, MYCL1, UT5320, and L17686 in colorectal cancer. New peaks or change of peak heights in cancer specimens indicate alteration of the repeat length compared to matching normal tissue.

**Figure 2 cancers-13-03529-f002:**
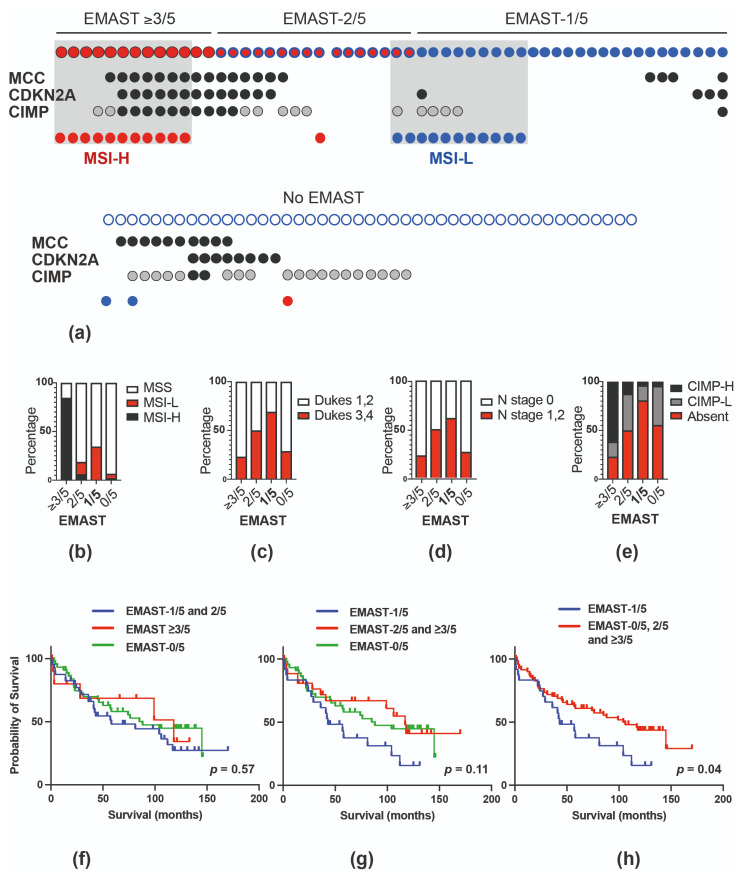
(**a)** Presence of CIMP-H, *CDKN2A* and *MCC* gene methylation (black), CIMP-L (grey), MSI-H (red) and MSI-L (blue) in 100 colorectal cancers grouped according to EMAST-≥3/5, 2/5, 1/5 or 0/5 status; distribution of MSI status (**b**), Dukes stages (**c**) N stages (**d**) and **(e)** CIMP status in colorectal cancers grouped according to EMAST-≥3/5, 2/5, 1/5 or 0/5 status (proportions taken from Table 1). Overall survival in patients grouped according to EMAST-positive (1–3 markers out of 5) or EMAST-negative tumours (**f,g**). Overall survival in patients with a single EMAST marker (EMAST-1/5) compared with all other patients (**h**).

**Figure 3 cancers-13-03529-f003:**
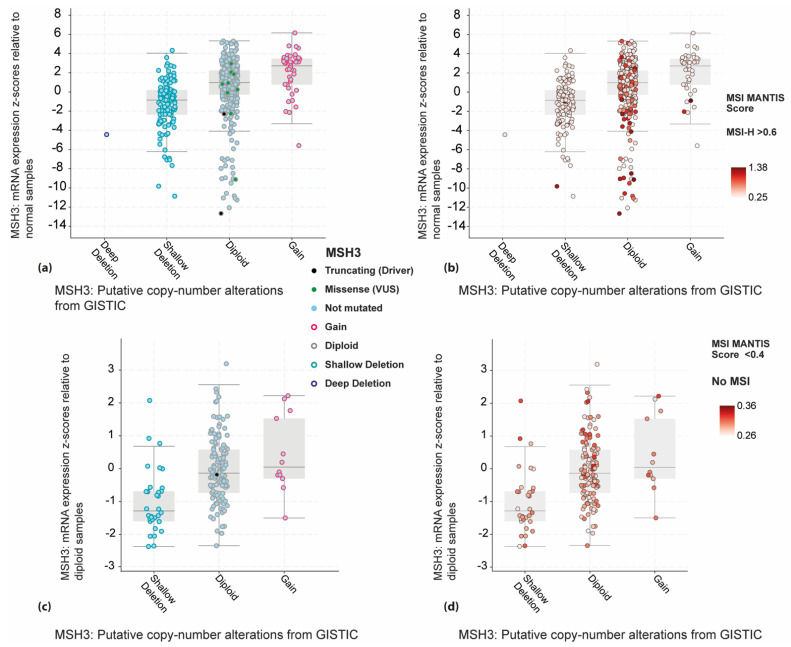
Genomic data from 524 colorectal and 168 pancreatic adenocarcinomas that were profiled for *MSH3* gene mutations, copy number alterations, mRNA expression and MSI MANTIS score in the TCGA PanCancer Atlas [32,33,34,35]. (**a**) Colorectal cancers have *MSH3* truncating mutation in 0.4% (2/524), one copy deletion in 24% (128/524), deep deletion in 0.2% (1/524), copy gain in 8% (41/524) and mRNA expression z-score (relative to normal samples) less than 2 in 17% (87/524) of the patients. The z-score for each tumour specimen indicates the number of standard deviations away from the mean expression in the reference population. VUS = variants of uncertain significance; shallow deletion = shallow loss, possibly a heterozygous deletion; deep deletion = deep loss, possibly a homozygous deletion; gain = low-level gain, a few additional copies. GISTIC = Genomic Identification of Significant Targets in Cancer. (**b**) MSI-H (MSI MANTIS score > 0.6 [35]) is found in 12% (64/522) and is mostly confined to MSH3 diploid colorectal cancers. (**c**) Pancreatic carcinomas have *MSH3* truncating mutation in 0.6% (1/168), one copy deletion in 19% (32/168) and copy gain in 7% (12/168) of the patients. **(d)** No MSI-H is found in pancreatic carcinoma, with MSI MANTIS score < 0.4 in all patients.

**Table 1 cancers-13-03529-t001:** Association of EMAST with clinicopathological features and molecular markers in colorectal cancer from cohort 1.

Tumour Feature	Variable	EMAST- ≥3/5 *n* = 13 (%)	EMAST- 2/5 *n* = 16 (%)	EMAST- 1/5 *n* = 26 (%)	No EMAST *n* = 45 (%)	*p ***
**Microsatellite** **instability**	MSS	2 (3)	13 (17)	17 (23)	42 (57)	**<0.0001**
	MSI-L	0 (0)	2 (15)	9 (70)	2 (15)	
	MSI-H	11 (84)	1 (8)	0 (0)	1 (8)	
**CIMP**	Absent	3 (5)	8 (14)	21 (37)	25 (44)	**<0.0001**
	CIMP-L	2 (7)	6 (20)	4 (13)	18 (60)	
	CIMP-H	8 (62)	2 (15)	1 (8)	2 (15)	
***CDKN2A* (p16) methylation**	Absent	5 (7)	11 (15)	22 (29)	37 (49)	**0.007**
	Present	8 (32)	5 (20)	4 (16)	8 (32)	
***MCC *** **methylation**	Absent	4 (6)	9 (13)	22 (31)	35 (50)	**0.002**
	Present	9 (31)	6 (21)	4 (14)	10 (34)	
***CDKN2A* & *MCC* methylation**	Absent	5(5)	11 (13)	25 (31)	41 (51)	**<0.00001**
	Present	8 (44)	5* (28)	1 * (6)	4 (22)	
**Dukes stage**	1,2	10 (17)	8 (14)	8 (14)	32 (55)	**0.028**
	3	2 (7)	6 (21)	11 (39)	9 (32)	
	4	1 (7)	2 (14)	7 (50)	4 (29)	
**T stage**	1,2	2 (7)	5 (19)	6 (22)	14 (52)	0.654
	3,4	11 (15)	11 (15)	20 (27)	31 (43)	
**N stage**	0	10 (16)	8 (13)	10 (16)	33 (54)	**0.036**
	1	1 (5)	4 (19)	7 (33)	9 (43)	
	2	2 (11)	4 (22)	9 (50)	3 (17)	
**M stage**	0	12 (14)	14 (16)	20 (23)	41 (47)	0.343
	1	1 (8)	2 (15)	6 (46)	4 (31)	
**Tumour site**	Proximal	11 (31)	6 (17)	7 (19)	12 (33)	**0.004**
	Distal	1 (3)	8 (23)	9 (26)	17 (48)	
	Rectum	1 (3)	2 (7)	10 (34)	16 (55)	
**Tumour size** **(median = 47)**	<47 mm	2 (4)	9 (19)	8 (17)	29 (60)	**0.002**
	≥47 mm	11 (22)	7 (14)	17 (35)	14 (29)	
**Sex**	Female	6 (12)	7 (14)	16 (33)	20 (41)	0.526
	Male	7 (14)	9 (17)	10 (20)	25 (49)	
**Differentiation**	Well/ Moderate	9 (10)	14 (16)	23 (27)	40 (47)	0.636
	Poor	3 (23)	2 (15)	3 (23)	5 (38)	

* EMAST-2/5 compared with EMAST-1/5, *p* = 0.023. ** Significant *p* values shown in bold.

**Table 2 cancers-13-03529-t002:** Association of MSI-H and MSI-L with clinicopathological features and molecular markers in sporadic colorectal cancer (cohort 1).

Tumour Feature	Variable	MSI-H *n* = 13 (%)	MSI-L *n* = 13 (%)	MSS *n* = 74 (%)	*p **
**Number of** **EMAST ** **markers**	0	1 (2)	2 (4)	42 (93)	**<0.0001**
	1–2	1 (2)	11 (26)	30 (71)	**-**
	3–5	11 (85)	0 (0)	2 (15)	-
**CIMP**	Absent	4 (7)	10 (18)	43 (75)	**0.003**
	CIMP-L	3 (10)	3 (10)	24 (80)	**-**
	CIMP-H	6 (46)	0 (0)	7 (54)	-
***CDKN2A* (p16) ** **methylation**	Absent	7 (9)	12 (16)	56 (75)	**0.014**
	Present	6 (24)	1 (4)	18 (72)	-
***MCC*** **methylation**	Absent	6 (8)	11 (16)	53 (76)	**0.004**
	Present	7 (24)	1 (3)	21 (72)	-
***CDKN2A* & *MCC *** **methylation**	Absent	7 (8)	13 (16)	62 (76)	**0.007**
	Present	6 (33)	0 (0)	12 (67)	-
**Dukes stage**	1.2	11 (19)	2 (3)	45 (78)	**0.007**
	3	2 (7)	7 (25)	19 (68)	-
	4	0 (0)	4 (29)	10 (71)	-
**T stage**	1,2	3 (11)	4 (15)	20 (74)	0.907
	3,4	10 (14)	9 (12)	54 (74)	-
**N stage**	0	11 (18)	4 (7)	46 (75)	**0.019**
	1,2	2 (5)	9 (23)	28 (72)	-
**M stage**	0	13 (15)	10 (11)	64 (74)	0.190
	1	0 (0)	3 (23)	10 (77)	-
**Tumour site**	Proximal	10 (28)	1 (3)	25 (69)	**0.006**
	Distal	2 (6)	7 (20)	26 (74)	-
	Rectum	1 (3)	5 (17)	23 (79)	-
**Tumour size**	<47 mm	4 (8)	8 (17)	36 (75)	0.270
	≥47 mm	9 (18)	5 (10)	35 (71)	-
**Sex**	Female	6 (12)	6 (12)	37 (76)	0.944
	Male	7 (14)	7 (14)	37 (72)	-
**Differentiation**	Well/ Moderate	10 (12)	12 (14)	64 (74)	0.787
	Poor	2 (15)	1 (8)	10 (77)	-

* Significant *p* values shown in bold.

**Table 3 cancers-13-03529-t003:** Association of *MSH3* mRNA expression with MSI, EMAST and gene promoter methylation in sporadic colorectal cancers (cohort 1). Tumour *MSH3* expression is shown relative to matched normal tissue and grouped according to median expression (0.63).

Tumour Feature	Variable	*MSH3* mRNA Expression <0.63 *n* = 48 (%)	*MSH3* mRNA Expression ≥0.63 *n* = 48 (%)	*p **
**Microsatellite** **instability**	**H**	10 (77)	3 (23)	**0.006**
	**L**	2 (15)	11 (85)	-
	**S**	36 (51)	34 (49)	-
**EMAST**	**≥3/5**	11 (85)	2 (15)	**0.011**
	**1–2/5**	21 (51)	20 (49)	-
	**0/5**	16 (38)	26 (62)	-
**CIMP**	**H**	11 (85)	2 (15)	**0.006**
	**L**	17 (57)	13 (43)	-
	**Absent**	20 (38)	33 (62)	-
***CDKN2A* (p16) ** **methylation**	**Yes**	16 (67)	8 (33)	0.098
	**No**	32 (44)	40 (56)	-
***MCC*** **methylation**	**Yes**	23 (82)	5 (18)	**0.0001**
	**No**	25 (37)	42 (63)	-

* Significant *p* values shown in bold.

## Data Availability

Publicly available TCGA PanCancer Atlas datasets were analysed in this study. The colorectal data can be found at https://www.cbioportal.org/results/plots?cancer_study_list=coadread_tcga_pan_can_atlas (accessed on 30 June 2021). Other datasets generated in this study are available from the corresponding author on reasonable request.

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
