# Peer review of "Tetranucleotide and Low Microsatellite Instability Are Inversely Associated with the CpG Island Methylator Phenotype in Colorectal Cancer"

_cancers, 2021, doi:10.3390/cancers13143529_

Round 1
Reviewer 1 Report
This manuscript describes the results of a study examining the relations between EMAST/MSI-L with MSI-H as well as EMAST/MSI-L with CIMP-H. Overall, it is very well-written and easy to follow/understand. The experiments were well designed and presented. There are only 5 points this reviewer is asking the authors to edit/clarify/answer. They are as following:
- In the "Abstract," the authors stated "Tumours with high microsatellite instability (MSI-H) display EMAST by default...." (line 36) This is not exactly correct. The correct statement should be "(very) high percentage of MSI-H tumors also exhibit EMAST." The authors' own data showed one MSI-H sample that did not show EMAST (Fig 2).
- In the "Result 3.1, authors noted that L17686 was unstable in the majority of the colitis-associated and sporadic pancreatic carcinomas, in sharp contrast to the sporadic CRC where 5 markers were evenly distributed (line 174-176). Inflammation plays a critical role in the tumorigenesis of pancreatic cancer, and IL-6 was found to significantly affect the progression of pancreatic cancers. This may suggest that L17686 could be a marker that is particular sensitive to inflammation induced EMAST/MSI-L. Please make a note of this point in the manuscript.
- Authors noted in the "Materials and Methods" that the "normal" controls for colitis-associated CRC were "non-carcinoma inflamed tissues, which were in fact diseased tissues that have been repeatedly exposed to highly inflammatory conditions (i.e. could be EMAST positive). This raised the possibility that the EMAST rate may have been higher than 62% (8/13) for this group. Please make a note of this point.
- In the "Result 3.6," authors stated that there are 24% (128/524) of samples had one copy of MSH3 deleted. However, only 17% (87/524) showed reduction of MSH3 mRNA level. Please explain. Additionally, 8% (41/524) showed gain of copies of MSH3. Do these samples have up-regulation of MSH3 mRNA levels?
- In the "Discussion" (lines332-334), authors proposed that EMAST may be a valuable predictive marker for cancers where MSI-H is rare. In fact, the most important point is to eliminate MSH-I samples, i.e. focus on MSH3-assoicated EMAST/MSI-L. EMAST/MSI-L could still be a valuable predictive marker in CRC, although 10-15% of sporadic CRC are MSI-H.
Author Response
In the "Abstract," the authors stated "Tumours with high microsatellite instability (MSI-H) display EMAST by default...." (line 36) This is not exactly correct. The correct statement should be "(very) high percentage of MSI-H tumors also exhibit EMAST." The authors' own data showed one MSI-H sample that did not show EMAST (Fig 2).
The abstract has been corrected as requested.
In the "Result 3.1, authors noted that L17686 was unstable in the majority of the colitis-associated and sporadic pancreatic carcinomas, in sharp contrast to the sporadic CRC where 5 markers were evenly distributed (line 174-176). Inflammation plays a critical role in the tumorigenesis of pancreatic cancer, and IL-6 was found to significantly affect the progression of pancreatic cancers. This may suggest that L17686 could be a marker that is particular sensitive to inflammation induced EMAST/MSI-L. Please make a note of this point in the manuscript.
We agree that this is an important point and has been added to the discussion (page 13).
Authors noted in the "Materials and Methods" that the "normal" controls for colitis-associated CRC were "non-carcinoma inflamed tissues, which were in fact diseased tissues that have been repeatedly exposed to highly inflammatory conditions (i.e. could be EMAST positive). This raised the possibility that the EMAST rate may have been higher than 62% (8/13) for this group. Please make a note of this point.
This has also been added to the discussion (page 13).
In the "Result 3.6," authors stated that there are 24% (128/524) of samples had one copy of MSH3 deleted. However, only 17% (87/524) showed reduction of MSH3 mRNA level. Please explain. Additionally, 8% (41/524) showed gain of copies of MSH3. Do these samples have up-regulation of MSH3 mRNA levels?
The Z-score of tumour MSH3 expression (Figure 3 graphs Y-axis) indicates the number of standard deviations away from the mean expression of normal tissue specimens. Therefore, Z-score of less than -2 indicates reduced mRNA expression and is seen in 17% of the tumours. A proportion (63%, 26/41) of the tumours with gain of copy number have a Z-score >2 (up-regulated). The corresponding frequency of up-regulation is 26% in the whole cohort (138/524).
Our conclusion of this data is that MSH3 mutations or deep deletion only explain a small proportion of EMAST in non-MSI-H colorectal or pancreatic carcinomas. Reduced MSH3 mRNA expression is found in diploid and copy-deleted/copy-gain tumours. It correlates with EMAST³3 but not with EMAST-1/2 in our data. These findings strengthen the view that MSH3 gene level alterations are not the main cause of EMAST.
We have amended the Discussion (second and third paragraphs) regarding MSH3 mRNA expression and the possible significance of MSH3 copy number alterations in relation to EMAST.
In the "Discussion" (lines332-334), authors proposed that EMAST may be a valuable predictive marker for cancers where MSI-H is rare. In fact, the most important point is to eliminate MSH-I samples, i.e. focus on MSH3-assoicated EMAST/MSI-L. EMAST/MSI-L could still be a valuable predictive marker in CRC, although 10-15% of sporadic CRC are MSI-H.
We have removed the statement “where MSI-H is rare”. The sentence now reads “MSH3-associated EMAST/MSI-L is a potentially valuable predictive biomarker for colorectal and other cancers, e.g. pancreatic and non-small cell lung carcinomas.”
We thank the reviewer for these valuable suggestions.
Reviewer 2 Report
This study is designed to assess in colorectal cancer the association of EMAST with several clinicopathological parameters including MSI and CIMP. The analysis of the methylation of selected genes known to be involved in this cancer were also analyzed. This is a well written manuscript that describes new features of tetranucleotide instability in colorectal cancer. The reported information is with great interest for clinicians.
The main topic of this manuscript is to assess in colon cancer the existence or not of a correlation or association between microsatellite instability and methylation of genes that were selected for the study by the authors based on their implication in this cancer. The correlation to several clinical parameters were also investigated such as the tumor grade, size and location of the tumor.
The study is not 100% original but validated some previously reported observation and added more information in the field by giving some explanation. Such as :
“ In this study we used five tetranucleotide repeat markers, D9S242, D20S82, L17686, 341 UT5320 and MYCL1 that were selected from a previous publication.[6] Two of these mark-342 ers, L17686, UT5320, have not been commonly used in other recent studies on EMAST. 343 Here we detected a higher frequency of EMAST in pancreatic cancer (74%) than previ-344 ously reported (45%),[42] which was due to the L17686 marker. Positivity of a single 345 marker was common in pancreatic cancers (48%, 15/31) using the EMAST marker panel 346 but less frequent (10%, 3/31) using the MSI marker panel. This highlights the possible 347 Cancers 2021, 13, x FOR PEER REVIEW 12 of 14 differences between cancer types and the need to standardise EMAST marker panels and 348 the nomenclature, possibly incorporating MSI and EMAST markers as was suggested by 349 Garcia et al[12] and Raeker et al[13]. “
It will be more informative if the authors added some information about the treatment response of the selected patients to immunotherapy. Indeed, some MSI colon cancer patients respond well to this treatment compared to MSS patients. Gene methylation/ Is it also involved? can we identify the involved methylated genes in the responder and none responder patients.
The conclusion can be improved by discussing the limited study that is based on the choice of specific selected genes to study their methylation. And also, how the EMAST marker can be combined to response to immunotherapy can be more efficient for patients stratification and treatment.
- It will be appreciated if available to add in the tables 1-3 some information related to response to treatment ( immunotherapy or chemotherapy)
- The size of then legend and the resolution of Figure 3 need to be improved
- Statistics ( SD, how many experiments etc.. ) need to shown in figures 2b-2e
Author Response
It will be more informative if the authors added some information about the treatment response of the selected patients to immunotherapy. Indeed, some MSI colon cancer patients respond well to this treatment compared to MSS patients. Gene methylation/ Is it also involved? can we identify the involved methylated genes in the responder and none responder patients.
The conclusion can be improved by discussing the limited study that is based on the choice of specific selected genes to study their methylation. And also, how the EMAST marker can be combined to response to immunotherapy can be more efficient for patients stratification and treatment.
- It will be appreciated if available to add in the tables 1-3 some information related to response to treatment ( immunotherapy or chemotherapy)
We agree that it is very important to investigate the role of EMAST in therapy responsiveness. However, all cancer patients in this retrospective cohort underwent surgery in 2000-2003 before immunotherapy was available. Therefore, we are unable to add data about immunotherapy responses. As for chemotherapy, this was given as adjuvant therapy for only 34 of the 100 patients.
Page 2, line 103. An explanation has been added in Materials and Methods that the specimens were “obtained from cancer resections (2000-2003). Thirty four percent of the patients received subsequent chemotherapy.”
- The size of then legend and the resolution of Figure 3 need to be improved
We have supplied an alternative presentation of Figure 3. The original high-resolution image is submitted separately.
- Statistics ( SD, how many experiments etc.. ) need to shown in figures 2b-2e
Figure 2b-2e graphs represent the proportions of the tumour features taken from Table 1. A clarification has been added to the figure legend. Also the order of the panels 2b-2e has been changed.
We thank the reviewer for these valuable suggestions.
Reviewer 3 Report
The study investigates associations between the tetranucleotide and classical microsatellite instability (EMAST and MSI, respectively) and CpG island methylator phenotype (CIMP) in colorectal cancer. The results are interesting and could be useful for further investigations. However, the manuscript is hard to read and lacks some key references that are necessary for proper interpretation and discussion. Major revision is necessary to improve the quality of data presentation and highlight the novelty of presented findings.
- Authors did not cite several recent studies that investigated associations EMAST and MSI in CRC tumors using very similar methods/marker panels. Authors shall carefully revise the Discussion and References in order to compare their findings with previously published results; e.g. Kondelin et al 2021 (DOI: 10.1002/gcc.22941), Pirini et al 2020 (doi:10.3390/ijms21103532), Park et al 2020 (DOI: 10.1002/jso.26157) and other potentially-relevant studies such as doi.org/10.1371/journal.pone.0197681 or doi:10.1371/journal.pone.0124538. Reviewer believes that at least some of those articles shall be included in the Discussion and it will help to prove and highlight the novelty of the submitted manuscript.
- Statements in lines 320-321 and 334-336 - add specific reference.
- The Results are presented in a diffuse way and it is very difficult to follow. Data presentation is a bit chaotic and not supported by the proper referencing of Figures and/or tables. Authors shall carefully revise the structure of paragraphs/subsections, shorten the sentences and add proper references (figure, table etc.) for each statement.
- The study will be much more coherent, concise and easier to follow if the results for pancreatic cancer samples (cohort 4) are excluded from the article. There is no information regarding the pancreatic cancer samples in the abstract nor introduction/aims of the study. Moreover, this issue is poorly discussed later.
- Methodology of TCGA dataset analysis shall be described in the methods chapter.
- As stated in lines 100-101 the authors collected follow-up data for the patients from cohort 1. Did authors utilize those data in any way? The survival analysis could improve the scientific soundness of the study and allow for better data interpretation. Similarly, the lymph node samples were collected (line 98), but it is unclear whether they have been analyzed for EMAST/MSI/CIMP?
- line 121 “Analysis of MSI and EMAST” shall be revised as “Analysis of EMAST and MSI” to follow uniform order methods/results presentation. In addition line 138, “Capillary electrophoresis…” shall be transferred to the new paragraph since it refers (?) to both EMAST and MSI PCR products.
- Minor remark: please check in-text citations for proper positioning in relation to the punctation markers.
Author Response
Authors did not cite several recent studies that investigated associations EMAST and MSI in CRC tumors using very similar methods/marker panels. Authors shall carefully revise the Discussion and References in order to compare their findings with previously published results; e.g. Kondelin et al 2021 (DOI: 10.1002/gcc.22941), Pirini et al 2020 (doi:10.3390/ijms21103532), Park et al 2020 (DOI: 10.1002/jso.26157) and other potentially-relevant studies such as doi.org/10.1371/journal.pone.0197681 or doi:10.1371/journal.pone.0124538. Reviewer believes that at least some of those articles shall be included in the Discussion and it will help to prove and highlight the novelty of the submitted manuscript.
Thank you for bringing these recent studies to our attention. We have cited three of the papers in the discussion (Kondelin, Pirini, Venderbosch). Their conclusions are not directly comparable with our study because their definition of EMAST requires instability in at least 2/5 tetranucleotide markers. This definition results in a significant overlap with MSI-H. It also does not include the EMAST-1/5 group, which we found is associated with lymph node metastasis and lack of CIMP.
Only the frequency of EMAST-1/5 can be inferred from the data presented in two of the papers:
Inferred EMAST-1/5 frequencies are 23% (9/40; Kondelin) and 28% (26/94; Venderbosch) of MSS tumours. In our study EMAST-1/5 frequency was comparable (26% of the total cohort, 23% of MSS tumours or 30% of MSS/MSI-L tumours).
The inferred frequencies have been included in the discussion. We have also highlighted the definition of EMAST and our novel findings regarding EMAST-1/5.
Statements in lines 320-321 and 334-336 - add specific reference.
Two references, Weisenberger et al 2006 and Dietlein et al 2014 have been added.
The Results are presented in a diffuse way and it is very difficult to follow. Data presentation is a bit chaotic and not supported by the proper referencing of Figures and/or tables. Authors shall carefully revise the structure of paragraphs/subsections, shorten the sentences and add proper references (figure, table etc.) for each statement.
We thank the reviewer for pointing out these issues. We have carefully revised results sections 3.2, 3.3 and 3.4 and hope that the data presentation is now easier to follow.
The study will be much more coherent, concise and easier to follow if the results for pancreatic cancer samples (cohort 4) are excluded from the article. There is no information regarding the pancreatic cancer samples in the abstract nor introduction/aims of the study. Moreover, this issue is poorly discussed later.
Discussion has been added on pancreatic cancer and colitis-associated cancers on page 13. Unfortunately we were not able obtain enough DNA to conduct gene methylation analysis of the pancreatic and colitis-associated cancers and this limits the discussion. However, on balance we would like to keep the EMAST data on pancreatic cancers in the manuscript because of the interesting differences in the performance of this EMAST marker panel compared to sporadic colorectal cancers. This was also highlighted by both Reviewer 1 and 2 as particularly informative.
Methodology of TCGA dataset analysis shall be described in the methods chapter.
A paragraph has been added in the end of Materials and Methods.
As stated in lines 100-101 the authors collected follow-up data for the patients from cohort 1. Did authors utilize those data in any way? The survival analysis could improve the scientific soundness of the study and allow for better data interpretation. Similarly, the lymph node samples were collected (line 98), but it is unclear whether they have been analyzed for EMAST/MSI/CIMP?
We have added survival analysis in section 3.3.
We apologise for our oversight of mentioning that DNA was extracted from any excised lymph nodes. This has now been deleted (page 2, line 103) because these specimens were not available for EMAST/MSI/CIMP analysis.
line 121 “Analysis of MSI and EMAST” shall be revised as “Analysis of EMAST and MSI” to follow uniform order methods/results presentation. In addition line 138, “Capillary electrophoresis…” shall be transferred to the new paragraph since it refers (?) to both EMAST and MSI PCR products.
These two points have been corrected in the manuscript (page 3).
Minor remark: please check in-text citations for proper positioning in relation to the punctation markers.
This has been corrected throughout the manuscript.
We thank the reviewer for these valuable suggestions.
Round 2
Reviewer 3 Report
The revised manuscript has been considerably improved for publication. Authors properly answered reviewers comments. I recommend to accept the MS for publication.